# OpenReview forum: "Mitigating Spurious Correlation via Distributionally Robust Learning with Hierarchical Ambiguity Sets"
_ICLR.cc/2026/Conference — ICLR 2026 Poster_

### Official Review · Reviewer_tVrP · 2025-10-30

**Soundness:** 3
**Presentation:** 3
**Contribution:** 3
**Rating:** 6
**Confidence:** 4

**Summary:**

The authors propose a hierarchical uncertainty set approach that takes care of inter-group and intra-group shifts based on robust optimization. Their method is effective in improving the worst group accuracies under certain data shifts scenarios, especially those with shifts in distribution beyond just changing group proportions. They have evaluated their approach on multiple datasets and obtain small improvements against competing approaches.

**Strengths:**

- The proposed method is effective in improving the worst group accuracy in handling distribution shifts, compared to GroupDRO and competing methods on a variety of datasets including CMNIST, Waterbirds, and CelebA.

- The main idea of the method is clear and well-presented: in addition to uncertainty in shifts in group proportions, add a second layer of uncertainty in the within-group samples.

**Weaknesses:**

- Compared to vanilla GroupDRO, the improvement is relatively small (Table 2) for the unshifted case.

- As with other forms of robust optimization that handles uncertainty in the samples, there is a tradeoff between optimizing for average and worst-case accuracies. Although we could obtain better worst case accuracies with the authors' method, it suffers worse average accuracies compared to GroupDRO in Table 1. It could be difficult to determine this tradeoff in practice.

**Questions:**

- How do the authors determine the size of the within-group ambiguity set? This could be a difficult parameter to tune.

---

> ### Author Response · Authors · 2025-11-20
>
> We thank the reviewer for the thoughtful and detailed questions. The comments helped us clarify several conceptual points, and we address each of them below.
>
> ---
>
> **W1.**
> > Compared to vanilla GroupDRO, the improvement is relatively small (Table 2) for the unshifted case.
>
> We appreciate the reviewer’s observation. Our formulation explicitly enlarges the ambiguity set to account for a broader range of distributional uncertainty (as illustrated in Fig.1). Because the unshifted case does not exhibit the types of distributional changes our method is designed to address, large improvements in this setting are not necessarily expected.
>
> However, spurious-correlation benchmarks possess a distinctive structural property: they include a highly underrepresented minority group. From a statistical perspective, when a group constitutes only a small fraction of the training data, its empirical distribution provides a weak approximation to the underlying population distribution, introducing intrinsic uncertainty even in the absence of an explicit shift. Our method, which optimizes over a richer ambiguity set, is naturally suited to mitigate this limited-support uncertainty.
>
> This interpretation aligns with the empirical results. In CMNIST, the minority group accounts for about 10% of the training set, yielding a comparatively large number of samples for that group and thus a relatively reliable empirical approximation of its distribution; correspondingly, the improvement over vanilla GroupDRO is modest. In contrast, in Waterbirds and CelebA, the minority groups represent only about 1% of the training data, so each minority group is represented by far fewer samples, making their empirical distributions substantially more fragile. For Waterbirds, Table 2 shows limited improvement using the original labels, but Table 3 demonstrates that once previously documented mislabeling issues are corrected [1], our method yields a much larger benefit even under the unshifted case. CelebA, where the minority group is represented by a very small number of samples, similarly shows a more pronounced improvement.
>
> Overall, while dramatic gains are not always expected in fully unshifted settings, the inherent statistical fragility of minority groups in spurious-correlation benchmarks enables our approach to deliver clear and meaningful improvements over vanilla GroupDRO—precisely in the regimes where robustness to limited-support uncertainty is most critical.
>
> ---
>
> **W2.**
> > As with other forms of robust optimization that handles uncertainty in the samples, there is a tradeoff between optimizing for average and worst-case accuracies. Although we could obtain better worst case accuracies with the authors' method, it suffers worse average accuracies compared to GroupDRO in Table 1. It could be difficult to determine this tradeoff in practice.
>
> We thank the reviewer for raising this important point. We would first like to clarify how the "average accuracy" reported in Table 1 is defined. Following common practice in the spurious-correlation literature, we report the training-weighted average accuracy, i.e., the overall accuracy under the empirical group proportions of the training data. Under this metric, our method already matches or exceeds GroupDRO on two of the three benchmarks: for CMNIST, our approach improves both average and worst-group accuracies, and for Waterbirds the average accuracies of GroupDRO and our method are nearly identical. The only dataset where we observe a noticeable drop in training-weighted average accuracy is CelebA.
>
> CelebA, however, is an extreme case in terms of group imbalance: the minority group constitutes less than 1% of the training data, so the training-weighted average accuracy is almost completely dominated by the majority group. In such a regime, even a substantial gain in worst-group performance has only a negligible effect on this particular average metric. To better reflect scenarios where one might care about all groups more evenly, we also computed the average accuracy under a hypothetical uniform test mixture over groups. On CelebA, this yields
> - **GroupDRO** : $83.9 \pm 0.7$
> - **Ours** : $86.6 \pm 0.7$
>
> showing that, when evaluated under equal group proportions, our method improves not only the worst-group accuracy but also the overall average accuracy.
>
> In summary, while the robustness-accuracy tradeoff is inherent, its manifestation depends strongly on the evaluation mixture. Under the conventional training-weighted metric, the degradation in average accuracy is small and limited to the most severely imbalanced dataset, and under a more balanced group mixture, our method in fact improves both average and worst-group accuracies. These observations suggest that, in practical scenarios where minority-group performance is important or where test-time group proportions may differ from the training distribution, the robustness–accuracy tradeoff remains moderate and, in fact, favorable for our method.

---

> ### Author Response · Authors · 2025-11-20
>
> **Q1.**
> > How do the authors determine the size of the within-group ambiguity set? This could be a difficult parameter to tune.
>
> We thank the reviewer for this question. As is common in DRO and adversarial training, the size of the ambiguity set (or perturbation radius) is a nontrivial hyperparameter to choose, and selecting it in a principled manner is generally challenging. Many existing works choose this radius heuristically or provide sensitivity analyses without prescribing a concrete tuning rule. In contrast, we propose a simple, data-driven procedure tailored to the intra-group shift setting considered in our work.
>
> Concretely, we parameterize the within-group radii as
> $$\epsilon_g = \epsilon / \sqrt{n_g},$$
> where $n_g$ is the size of group $g$ in the training data. This links the amount of within-group uncertainty directly to the statistical reliability of each group: smaller groups, whose empirical distributions are statistically less reliable, are assigned a larger radius, while larger groups receive a smaller one.
>
> The remaining global parameter $\epsilon$ is selected using the data-driven validation procedure described in Section 4.2 and Appendix G. Conceptually, if one had access to a validation set drawn from the shifted test distribution of interest, $\epsilon$ could be chosen by standard model selection to optimize performance under that specific shift. However, our goal is to adhere to the more realistic scenario in which the future shift is unknown and no such shifted validation data are available. We therefore propose a procedure that relies solely on the training data to construct synthetic intra-group shifts and select $\epsilon$ accordingly.
>
> We believe that, together, the $\epsilon_g = \epsilon / \sqrt{n_g}$ scaling and this validation strategy provide a simple, intuitive, and practically useful guideline for determining the size of the within-group ambiguity set.
>
> ---
>
> We sincerely appreciate the reviewer’s careful reading and insightful questions, which allowed us to clarify the motivations and assumptions underlying our framework.
>
> ---
>
> **Reference**
>
> [1] Asgari et al. Masktune: Mitigating spurious correlations by forcing to explore, NeurIPS, 2022.

---

### Official Review · Reviewer_GbgA · 2025-11-01

**Soundness:** 3
**Presentation:** 3
**Contribution:** 3
**Rating:** 6
**Confidence:** 3

**Summary:**

This paper addresses a limitation of existing methods for mitigating spurious correlations, such as GroupDRO, which are robust to shifts in group proportions (inter-group shifts) but fail when the data distribution within a minority group changes between training and testing (intra-group shifts). The authors propose a hierarchical distributionally robust optimization (DRO) framework that models both levels of uncertainty through a hierarchical ambiguity set. This set forces the model to be robust not only to the worst-case mixture of groups but also to the worst-case data variations within each group, formulated using a Wasserstein distance in a latent space. To validate their approach, they introduce new challenging benchmark datasets designed to exhibit these minority-group shifts, demonstrating that their method achieves stronger robustness where state-of-the-art methods fail, while also maintaining top performance on standard benchmarks.

**Strengths:**

This paper presents a well-motivated extension to distributionally robust optimization (DRO) for mitigating spurious correlations. It identifies and addresses a failure mode in existing robust learning methods: vulnerability to intra-group distribution shifts.

While prior work like Group DRO focuses on robustness to changing group proportions (inter-group shifts), this paper argues that the distribution within small minority groups can itself be non-stationary.

The proposed hierarchical ambiguity set is a theoretically sound solution that unifies robustness at both the inter- and intra-group levels. The authors introduce new, modified datasets specifically designed to induce the targeted minority-group shifts, providing a direct test of their hypothesis.

The motivation is made intuitive through clear diagrams (Fig. 1), the mathematical formulation is precise, and the results are analyzed thoroughly, including qualitative insights from Grad-CAM.

**Weaknesses:**

I've identified several gaps and issues:

The proof applies Lemma A.0.1 (from Staib & Jegelka 2017) but then immediately relaxes it:

$$E_{P_g}\left[\sup_{x: d(z(x), z(X)) \leq \epsilon_g} L(f_\theta^L(z(x)), Y)\right] \leq E_{P_g}\left[\sup_{z': d(z', z(X)) \leq \epsilon_g} L(f_\theta^L(z'), Y)\right]$$

- This inequality can be strict (not tight) when $z(\cdot)$ is not surjective onto the $\epsilon$-ball

Also proposition B.1 assumes:
- "the feature map $z(x)$ is **fixed** w.r.t. $\theta$"

But $z(x)$ is defined (equation 7) as:

$z(x) := f_\theta^{L-1}(f_\theta^{L-2}(\ldots f_\theta^1(x)))$

This is a direct contradiction - $z(x)$ clearly depends on $\theta$ through layers 1 to $L-1$. The convergence analysis is therefore invalid for the actual algorithm being run.

The cost function (page 5) sets:

$c((x,y), (x',y')) = \begin{cases} \|z(x) - z(x')\|, & \text{if } y = y' \\\\ \infty, & \text{otherwise} \end{cases}$

This means $W_p(P, Q) = \infty$ whenever marginal distributions of $Y$ differ.

- The paper claims "this definition does not cause any issues" because $G = (Y, A)$, but this severely restricts the ambiguity set - distributions can only shift within same-label groups

The paper doesn't compare against:
- Standard Wasserstein DRO methods (Kuhn et al. 2019, cited but not compared)

Also as a notation note, $P$ vs $\hat{P}$ usage is sometimes unclear.

**Questions:**

See weaknesses.

---

> ### Author Response · Authors · 2025-11-20
>
> We thank the reviewer for the careful and detailed reading of our manuscript and for the insightful comments. We address each point below.
>
> ---
>
> **W1.**
> > The proof applies Lemma A.0.1 (from Staib & Jegelka 2017) but then immediately relaxes it:
> $$\mathbb{E}_{P_g}\left[ \sup _{x :\|z(x)-z(X)\ | \le\epsilon_g}\mathcal{L}(f^\theta_L(z(x)),Y)\right] \le \mathbb{E} _{P_g}\left[\sup _{z':\|z'-z(X)\|\le\epsilon_g}\mathcal{L}(f^\theta_L(z'),Y)\right]$$
> This inequality can be strict (not tight) when $z(\cdot)$ is not surjective onto the $\epsilon$-ball.
>
> We thank the reviewer for raising this important point. While we agree that the inequality may be strict in general when the feature map $z(\cdot)$ is not surjective onto the $\epsilon_g$-ball, we believe that this phenomenon is negligible in practice for embedding distributions arising in modern deep networks. This is because it is widely assumed that the pushforward distribution of $X$ under the embedding $z(\cdot)$ possesses a positive Lebesgue density on, or in a neighborhood of, its effective support.
>
> Although raw images often concentrate near a low-dimensional manifold in the input space and therefore may not admit a Lebesgue density, the situation is different after they pass through several nonlinear layers. Empirically and conceptually, internal representations produced by modern deep networks behave as “thickened” manifolds and are typically modeled as distributions with nonzero volume in the ambient space. A notable example supporting this view is the widespread use of the Fréchet Inception Distance (FID) for evaluating generative models, which implicitly treats the distribution of the embedding as having a non-degenerate density in feature space.
>
> Formally proving the existence of a density for arbitrary deep embeddings is challenging. However, once we adopt this commonly held assumption, as is standard in practice, the relaxation introduced after Lemma A.0.1 becomes nearly tight. In such cases, the overlap between the $\epsilon_g$-ball and the image of $z(\cdot)$ is substantial, making the resulting upper bound a close approximation of the original hierarchical objective.
>
> We have added this clarification in Section 4.2 (highlighted in blue in the revised manuscript).
>
> ---
>
> **W2.**
>
> > Proposition B.1 assumes:
> the feature map $z(x)$ is fixed w.r.t. $\theta$.
> But $z(x)$ is defined (equation 7) as :
> $$
> z(x) := f_{L-1}^\theta \left( f_{L-2}^\theta \left( \ldots f_1^\theta(x) \right) \right).
> $$
> This is a direct contradiction - $z(x)$ clearly depends on
> $\theta$ through layers 1 to $L-1$. The convergence analysis is therefore invalid for the actual algorithm being run.
>
> We appreciate the reviewer’s observation. The reviewer is correct that there is a gap between our practical algorithm and Proposition B.1. Since establishing convergence to a global optimum is generally intractable without convexity, our theoretical analysis adopts the standard convexity assumption, which is typically formalized by treating the feature map $z(\cdot)$ as fixed during the convergence argument. This follows the common practice in theoretical analyses of adversarial training and distributionally robust optimization.
>
> We acknowledge the gap between the practical algorithm and the theoretical analysis and have added a clarification in Appendix B (highlighted in blue in the revised manuscript).
>
> ---
>
> **W3.**
>
> > The cost function (page 5) sets:
> $$
> c((x,y),(x',y')) =
> \\begin{cases}
>   \\|z(x) - z(x')\\| & \\text{if } y = y',\\\\
>   \\infty           & \\text{otherwise}.
> \\end{cases}
> $$
> This mean $W_p(P, Q) = \infty$ whenever marginal distributions of $Y$ differ. The paper claims “this definition does not cause any issues” because $G = (Y, A)$, but this severely restricts the ambiguity set - distributions can only shift within same-label groups.
>
> We believe there is some misunderstanding here. Since the group indicator $G$ is defined as the pair $(Y, A)$, each group contains data points with a single label $Y$. Consequently, the case distinction in the cost function—assigning infinity when the labels differ—is never activated within a group, and the value “$\infty$’’ can in fact be replaced by any sufficiently large constant without affecting the ambiguity set.
>
> In this sense, it would indeed be more natural to define the cost (and thus the Wasserstein distance) solely on the marginal distribution of $X$, rather than on the joint distribution of $(X, Y)$. We chose to maintain the joint-distribution formulation purely for notational consistency, because probability measures such as $P$ and $Q$ throughout the paper denote joint distributions.

---

> ### Author Response · Authors · 2025-11-20
>
> **W4.**
>
> > The paper doesn't compare against:
> Standard Wasserstein DRO methods (Kuhn et al. 2019, cited but not compared).
>
> We thank the reviewer for raising this suggestion. We have conducted additional experiments to compare our method against standard Wasserstein DRO methods. As anticipated from the structure of spurious-correlation benchmarks—and in particular under the minority group shift setting that our work focuses on—standard Wasserstein DRO performs poorly, for two main reasons.
>
> First, standard Wasserstein DRO models distributional uncertainty solely by allowing perturbations of the distribution within a single Wasserstein ball, but it does not incorporate any uncertainty over group proportions. In spurious-correlation settings, however, changes in group proportions constitute a key form of distribution shift: a minority group that is scarcely observed during training may appear far more frequently at test time. In contrast, our hierarchical formulation explicitly models uncertainty both at the mixture level and at the level of each group's conditional distribution, where the latter is captured through a Wasserstein ball around the empirical within-group distribution. This enables our method to directly address the kinds of shifts that arise in these benchmarks.
>
> Second, in spurious-correlation benchmarks, different groups have substantially different sample sizes. Our method is designed to reflect this by assigning a different within-group radius to each group ($\epsilon_g = \epsilon / \sqrt{n_g}$), so that smaller (statistically less reliable) groups are assigned a larger perturbation radius. Standard Wasserstein DRO uses a single global radius and therefore cannot adapt to group-specific statistical uncertainty, which is crucial in the minority-shift setting.
>
> For a fair comparison, we implemented standard Wasserstein DRO with the same latent-space cost function used in our method (ensuring both methods capture semantic variation), and matched all other training hyperparameters. We varied the Wasserstein radius over a wide range of values and report the best worst-group accuracy achieved by standard Wasserstein DRO across all choices. The results are:
>
> - **Waterbirds** : 40.2% (best over all radii), compared to ERM at 37.9%.
> - **CelebA** : 30.2% (best over all radii), compared to ERM at 19.8%.
>
> While standard Wasserstein DRO improves slightly over ERM, its best accuracies remain far below those of our method (93.7\% and 72.1\%, respectively), confirming that modeling only Wasserstein perturbations is insufficient to handle the complex distribution shifts present in spurious-correlation benchmarks.
>
> These findings support the necessity of the hierarchical perspective introduced in our framework: explicitly modeling uncertainty in group proportions and tailoring within-group radii to group sizes are both essential for achieving strong robustness in spurious-correlation settings.
>
> We have incorporated these additional experimental results and the corresponding performance–radius plots into Appendix I of the revised manuscript (highlighted in blue in the revised manuscript).
>
> ---
>
> **W5.**
>
> >Also as a notation note,
>  $P$ vs $\hat{P}$ usage is sometimes unclear.
>
> We appreciate the reviewer for raising this point. Could you kindly indicate which specific instances of $P$ or $\hat{P}$ were unclear? In our notation, $\hat P$ consistently denotes an estimator of $P$, and we would be happy to revise or clarify any usages that appeared ambiguous.
>
> ---
>
> We sincerely appreciate the reviewer’s thorough and constructive feedback. The comments have helped us improve the clarity, correctness, and presentation of the theoretical components of the paper.

---

> > ### Comment · Reviewer_GbgA · 2025-11-26
> >
> > I appreciate the authors’ thorough and comprehensive response. I have decided to raise my score.

---

### Official Review · Reviewer_wr5K · 2025-11-01

**Soundness:** 3
**Presentation:** 4
**Contribution:** 3
**Rating:** 8
**Confidence:** 3

**Summary:**

This paper proposed a robust optimization scheme with hierarchical distribution shifts. It optimizes the worst-case distribution from a mixture of shifted group distributions Eq.(5). As a concrete instance, the mixture coefficients can be arbitrary and the group drift is constrained within a Wasserstein ball as in Eq.(8). A practical algorithm is introduced with convergence guarantee. Experiments on common robust optimization benchmark datasets show that the algorithm can outperform several baselines in the robust optimization literature.

**Strengths:**

1. Writing and presentation are very clear and easy to follow

2. The algorithm is theoretically motivated, and a practical algorithm is introduced with a convergence guarantee.

3. The algorithm performs better than several baselines in terms of worst-case accuracy.

**Weaknesses:**

1. A hierarchical framework is introduced but with only one instantiation. It would be better to showcase the flexibility of the framework by using the $\rho$ in (5). More importantly, it is unclear why such a $\rho$ constraint is meaningful in real-world scenarios.

2. Similarly, only the Wasserstein distance is considered as the measure of distribution shift. It would be worthwhile to discuss other options like f-divergence, and whether they will be feasible in practice.

**Questions:**

What other options would be possible and practical for the general framework specified in (5)?

---

> ### Author Response · Authors · 2025-11-20
>
> We thank the reviewer for the thoughtful and constructive comments. Below we address each point in detail.
>
> ---
>
> **W1.**
> > A hierarchical framework is introduced but with only one instantiation. It would be better to showcase the flexibility of the framework by using the in (5). More importantly, it is unclear why such a $\rho$ constraint is meaningful in real-world scenarios.
>
> We appreciate the reviewer’s question. The purpose of Eq.(5) is to decompose distributional uncertainty into two levels, and the term $d_1(\beta, \alpha) \le \rho$ controls **how much the test-time group proportions may differ from the training mixture**. In practice, the choice of $\rho$ depends on prior knowledge: if group proportions are believed to remain stable, a finite and relatively small $\rho$ is appropriate; if such prior knowledge is unavailable, a larger value is needed. We therefore include this constraint to allow the framework to accommodate scenarios where such prior knowledge exists, even though our primary focus lies in settings where it does not.
>
> In the spurious-correlation setting, it is standard to assume no reliable prior about the test group proportions. Worst-group accuracy—used in nearly all prior work—can be viewed as evaluating performance under the extreme mixture in which the entire probability mass of the test distribution is concentrated on the worst-performing group. This corresponds to taking $\rho = \infty$, giving the optimizer full freedom to focus on the most underrepresented group.
>
> Thus, while our framework can incorporate finite $\rho$ when prior knowledge about group proportions exists, we intentionally set $\rho = \infty$ to match the prevailing “worst-group” robustness assumption and to ensure that the optimization meaningfully emphasizes the underrepresented group, which is essential for mitigating spurious correlations.
>
> We have added clarifications regarding this motivation and the practical interpretation of $\rho$ in Section 4.1 (highlighted in blue in the revised manuscript).
>
> ---
>
> **W2.**
> > Similarly, only the Wasserstein distance is considered as the measure of distribution shift. It would be worthwhile to discuss other options like f-divergence, and whether they will be feasible in practice.
>
> We thank the reviewer for highlighting this important point. While f-divergences are in principle feasible, they impose a structural restriction that makes them less suitable for our setting. For any f-divergence, the ambiguity set can only contain distributions that are absolutely continuous with respect to the empirical distribution. Since the empirical distribution has finite support, this implies that any candidate distribution must assign probability only on this same support. In practice, this means that the ambiguity set can only reweight the finite empirical instances—an especially severe limitation for minority groups with very limited support.
>
> In contrast, the Wasserstein distance does not impose absolute continuity and therefore allows support shifts, enabling the ambiguity set to include semantically plausible variations that may not appear in the training set. This is crucial for the type of intra-group shifts we consider. For example, in CelebA, the minority group (e.g., blond males) may include test-time examples wearing glasses, even though the training samples from this group do not include examples with glasses. Such unseen semantic variation cannot be modeled by reweighting the finite empirical support. Wasserstein-based ambiguity sets naturally accommodate this kind of semantic shift, while f-divergence–based sets cannot.
>
> For these reasons, although alternative discrepancy measures are theoretically possible, Wasserstein distance provides a more realistic modeling of intra-group distribution shifts in practice.
>
> We have added this clarification in Section 4.1 (highlighted in blue in the revised manuscript).

---

> ### Author Response · Authors · 2025-11-20
>
> **Q1.**
> > What other options would be possible and practical for the general framework specified in (5)?
>
> We appreciate the reviewer’s question. As discussed above, Eq.(5) provides a general hierarchical formulation that can flexibly incorporate different forms of distributional uncertainty. In this work, we instantiate the framework using choices of $d_1$ and $d_2$ that are most aligned with the spurious-correlation setting, where intra-group distribution shift naturally arises by modifying the standard train–test splits of widely used benchmarks. Because our paper is the first to explicitly formalize and study such intra-group shifts, we present the general ambiguity set and then instantiate it with the discrepancy measures that most directly capture this phenomenon.
>
> That said, exploring alternative instantiations of the framework can be an important research direction. For instance, although we use the Wasserstein distance for $d_2$ due to its strong theoretical foundations in DRO and its ability to model semantic support shifts, other discrepancy measures may be appropriate under different modeling assumptions. One promising candidate is the unbalanced Wasserstein distance, which has been studied extensively in recent years and blends the geometric structure of optimal transport with the mass-variation flexibility provided by KL-type divergence terms [1,2]. Such distances may be advantageous in settings where the empirical distribution contains outliers or where mass variation—rather than pure transport—better captures the underlying uncertainty.
>
> We view the systematic investigation of these alternatives as a promising avenue for future work that is complementary to the specific instantiation developed in this paper.
>
> ---
>
> We sincerely thank the reviewer for the detailed and constructive feedback, which has helped us improve both the clarity and presentation of the manuscript.
>
> ---
>
> **Reference**
>
> [1] Wang et al. Outlier-robust distributionally robust optimization via unbalanced optimal transport, NeurIPS, 2024.
>
> [2] Nguyen et al. On Unbalanced Optimal Transport: Gradient Methods, Sparsity and Approximation Error, JMLR, 2023.

---

### Author Response · Authors · 2025-12-02
**Summary of Reviewer Discussion**

Dear Area Chair,

We sincerely appreciate your efforts in evaluating our submission during this challenging review cycle. For your convenience, we provide below a concise summary of the reviewer discussion. For context, our work focuses on realistic minority-group distribution shifts—an important yet previously underexplored challenge in spurious-correlation research—which existing robust-learning methods consistently fail to handle.

All reviewer concerns—conceptual, theoretical, and experimental—were carefully addressed in the rebuttal and corresponding **revisions to the manuscript**. One reviewer (GbgA) increased their score from 6 to 8 after reading the responses. Although further exchanges with the remaining two reviewers were not possible, we believe that our responses adequately addressed all the questions they had raised.

---

**1. Reviewer wr5K**

**Initial rating : 8**

**Main concerns :**

- Clarification of the framework’s flexibility in Eq. (5) and the mixture-level constraint.
- Alternative discrepancy measures (e.g., $f$-divergences) and their feasibility.
- Possible alternative instantiations of the general framework in Eq. (5).


**Our response :**
We provided detailed explanations addressing all three points:

(1) clarified the flexibility of the hierarchical ambiguity set and added explicit discussion on the role of the mixture-level constraint (**Section 4.1**).

(2) expanded the discussion comparing Wasserstein ambiguity sets with $f$-divergence alternatives, explaining why the latter are structurally limited in our setting (**Section 4.1**).

(3) described feasible alternative instantiations of Eq. (5) and justified our choice as the most suitable for modeling intra-group shifts in spurious-correlation benchmarks.

---

**2. Reviewer GbgA**

**Initial rating : 6 (→ 8)**

**Main concerns :**

- Tightness of the relaxation following Lemma A.0.1.
- Assumption that the feature map is fixed in Proposition B.1.
- Cost function assigning $\infty$ across labels.
- Experimental comparison against standard Wasserstein DRO.
- Minor notational ambiguity regarding $P$ vs. $\hat{P}$.

**Our response :**
We addressed all technical concerns in detail:

(1) clarified the density assumptions commonly adopted for deep embeddings (**Section 4.2**).

(2) clarified the convexity-based assumption in Proposition B.1 (**Appendix B**).

(3) explained why the $\infty$ branch of the cost function is irrelevant under group-conditional shifts.

(4) added new experiments comparing against standard Wasserstein DRO, including performance–radius curves (**Appendix I**).

(5) clarified the notation where necessary.

---

**3. Reviewer tVrP**

**Initial rating : 6**

**Main concerns :**

- Modest gains over GroupDRO in the unshifted case.
- Possible robustness–accuracy tradeoff, based on the average accuracies in Table 1.
- How to select the within-group radius $\epsilon_g$.

**Our response :**
We addressed all of the reviewer’s concerns in detail:

(1) clarified that large gains are not expected in the fully unshifted setting, while explaining how the structural properties of spurious-correlation benchmarks—especially the extreme scarcity of minority-group samples—still allow our richer ambiguity set to yield meaningful improvements.

(2) explained that the robustness–accuracy tradeoff depends strongly on the assumed test-time group proportions, and showed that under a uniform group distribution our method improves both average and worst-group accuracy.

(3) detailed our practical, data-driven procedure for determining the group-specific radii ($\epsilon_g = \epsilon / \sqrt{n_g}$) and for selecting the global parameter $\epsilon$ (**Sections 4.2** and **Appendix G**).

---

We hope this summary assists the AC in efficiently understanding the review history and the completeness of the revisions.

---

### Meta-Review · Area_Chair_LkbL · 2026-01-07

**Summary:**

This paper introduces a hierarchical distributionally robust optimization (DRO) framework designed to mitigate spurious correlations by addressing both inter-group (shifts in group proportions) and intra-group (distributional changes within a group) uncertainties. By utilizing a hierarchical ambiguity set that combines group-level reweighting with Wasserstein DRO in a latent space, the method provides robustness to realistic minority group shifts where standard Group DRO often fails.

**Reviewer Concerns:**

Technical queries regarding the flexibility of the hierarchical framework, the choice of Wasserstein distance over f-divergences, and hyperparameter tuning were successfully addressed through detailed rebuttals and the inclusion of new baseline comparisons.

**Reviewer Scores:**

The final scores are 8, 8, and 6, reflecting a strong positive consensus and a successful score increase from Reviewer GbgA following the rebuttal.

---

### Decision · Program_Chairs · 2026-01-26

Accept (Poster)